# A High-Resolution Global Map of Giant Kelp (*Macrocystis pyrifera*) Forests and Intertidal Green Algae (*Ulvophyceae*) with Sentinel-2 Imagery

**Alejandra Mora-Soto** [1,*], **Mauricio Palacios** [2,3,4,5], **Erasmo C. Macaya** [5,6,7], **Iván Gómez** [2,5], **Pirjo Huovinen** [2,5], **Alejandro Pérez-Matus** [8], **Mary Young** [9], **Neil Golding** [10], **Martin Toro** [11], **Mohammad Yaqub** [12] and **Marc Macias-Fauria** [1]

1　Biogeosciences Group, School of Geography and the Environment, University of Oxford, Oxford OX1 3QY, UK; marc.maciasfauria@ouce.ox.ac.uk
2　Instituto de Ciencias Marinas y Limnológicas, Facultad de Ciencias, Universidad Austral de Chile, Campus Isla Teja s/n, Valdivia 5090000, Chile; mauricio.palacios@alumnos.uach.cl (M.P.); igomezo@uach.cl (I.G.); pirjo.huovinen@uach.cl (P.H.)
3　Facultad de Ciencias, Universidad de Magallanes, Punta Arenas 6210427, Chile
4　Programa de Doctorado en Biología Marina, Universidad Austral de Chile, Valdivia 5090000, Chile
5　Centro FONDAP de Investigación en Dinámica de Ecosistemas Marinos de Altas Latitudes (IDEAL), Valdivia 5090000, Chile; emacaya@oceanografia.udec.cl
6　Laboratorio de Estudios Algales (ALGALAB), Departamento de Oceanografía, Universidad de Concepción, Casilla 160-C, Concepción 4030000, Chile
7　Millenium Nucleus Ecology and Sustainable Management of Oceanic Islands (ESMOI), Coquimbo 1780000, Chile
8　Subtidal Ecology Laboratory, Departamento de Ecología, Estación Costera de Investigaciones Marinas, Pontificia Universidad Católica de Chile, Casilla 114-D, Santiago 8320000, Chile; aperez@bio.puc.cl
9　Centre for Integrative Ecology, Life and Environmental Sciences, Deakin University, Princes Hwy, Warrnambool 3280, Australia; mary.young@deakin.edu.au
10　South Atlantic Environmental Research Institute (SAERI), Stanley FIQQ 1ZZ, Falkland Islands; ngolding@saeri.ac.fk
11　Instituto de Geografía, Pontificia Universidad Católica de Chile, Macul, Santiago 782-0436, Chile; mtoro4@uc.cl
12　IT Services, University of Oxford, Oxford OX2 6NN, UK; mohammad.yaqub@ndcn.ox.ac.uk
*　Correspondence: alejandra.morasoto@ouce.ox.ac.uk; Tel.: +44-(0)-1865-285070

**Abstract:** Giant kelp (*Macrocystis pyrifera*) is the most widely distributed kelp species on the planet, constituting one of the richest and most productive ecosystems on Earth, but detailed information on its distribution is entirely missing in some marine ecoregions, especially in the high latitudes of the Southern Hemisphere. Here, we present an algorithm based on a series of filter thresholds to detect giant kelp employing Sentinel-2 imagery. Given the overlap between the reflectances of giant kelp and intertidal green algae (*Ulvophyceae*), the latter are also detected on shallow rocky intertidal areas. The kelp filter algorithm was applied separately to vegetation indices, the Floating Algae Index (FAI), the Normalised Difference Vegetation Index (NDVI), and a novel formula (the Kelp Difference, KD). Training data from previously surveyed kelp forests and other coastal and ocean features were used to identify reflectance threshold values. This procedure was validated with independent field data collected with UAV imagery at a high spatial resolution and point-georeferenced sites at a low spatial resolution. When comparing UAV with Sentinel data (high-resolution validation), an average overall accuracy $\geq 0.88$ and Cohen's kappa $\geq 0.64$ coefficients were found in all three indices for canopies reaching the surface with extensions greater than 1 hectare, with the KD showing the highest average kappa score (0.66). Measurements between previously surveyed georeferenced points and remotely-sensed kelp grid cells (low-resolution validation) showed that 66% of the georeferenced

points had grid cells indicating kelp presence within a linear distance of 300 m. We employed the KD in our kelp filter algorithm to estimate the global extent of giant kelp and intertidal green algae per marine ecoregion and province, producing a high-resolution global map of giant kelp and intertidal green algae, powered by Google Earth Engine.

**Keywords:** giant kelp; *Macrocystis pyrifera*; Google Earth Engine; UAV; Sentinel-2; *Ulvophyceae*

## 1. Introduction

Kelp forests provide a variety of ecosystem services: they protect and modify the substrate and surrounding water column and act as a habitat for several species, increasing the complexity of coastal trophic networks [1,2]. Although the term 'kelp' is used to refer to all large brown algae [3], the focus of this study is the commonly called giant kelp [3], *Macrocystis pyrifera* (Laminariales, Phaeophyceae), Earth's most widespread kelp species, which forms one of the most productive and diverse ecosystems on the planet [4–6]. *M. pyrifera* forests (giant kelp hereafter) can be found in the temperate and subpolar coastlines of South America, Western North America, South Africa, Australia, New Zealand, as well as the South Atlantic and the majority of the Sub-Antarctic islands [7–9]. Despite the importance of kelp forests as a global biome, kelp monitoring has mostly been implemented locally and generally without temporal replication, which obscures the dynamics in many marine regions [10]. Mapping kelp's global distribution and biomass dynamics is thus a necessary first step in understanding its contemporary distribution and identifying possible threats in a context of global change [11]. This information can contribute to the sustainable development agenda of the UN by 2030, specifically goal 14: "Conserve and sustainably use the oceans, seas and marine resources for sustainable development" [12].

Canopies (fronds) of giant kelp reach the surface of the ocean and can be easily seen from aerial platforms at low altitude [13]. However, it is difficult to clearly distinguish kelp canopies from open ocean water in satellite imagery because of factors like: sun glint, breaking waves, sediments, dissolved organic matter or phytoplankton blooms that also contribute to water reflectance [14]. Techniques and sensors used for kelp mapping have included aerial photogrammetry [11,15], Landsat Multispectral Scanner System (MSS) [16], Landsat Thematic Mapper (TM) [13,14], Landsat Operational Land Imager (OLI) [17], Sentinel-2 [18], Satellite Pour l'Observation de la Terre (SPOT) [19,20], and aircraft multispectral sensors with spatial resolutions ~2m [17,21]). Methodologies employed to detect kelp canopies using such imagery have been varied: from using thresholds of band brightness value ratios to indices such as the Normalised Difference Vegetation Index (NDVI; [13]) to methods such as unsupervised classification [19,20], supervised maximum likelihood classification [16,17,22], Principal Component Analysis (PCA; [13]), Spectral Mixture Analysis (SMA; [18,23]), and particularly Multiple Endmember Spectral Mixture Analysis (MESMA; [14,24]), which has been effective in studying different characteristics of kelp ecology, such as genetic differentiation within geographic clusters [25], long-term biomass trends [26], fish, invertebrate and algae assemblages in kelp communities [27], and biogeochemical [28] and physical [14] dynamics.

The above-mentioned methods have been applied at local to regional scales, e.g., in the Santa Barbara Channel (Southern California) or the Francisco Coloane Marine Park, Cape Horn archipelago and Diego Ramirez islands in southern Chile [29]. These studies were not scaled up to the global distribution due to the high operative costs of human hours required for pre-processing and classifying images, as well as providing pure endmember data, especially along extensive coastlines (but see [30]). Previous research using citizen science data found that at least four different people were required to identify and classify the same grid cell as kelp in order to make an optimal supervised classification [30]. In a similar way, Machine Learning methods, such as Random Forest or Classification and Regression Trees (CART) [31], require categorical variables to split grid cell values in a series of thresholds in a tree-like structure, predicting a categorical label for each cell [32]. In these classification

algorithms, data classes define the size of the tree. That is, with limited training data, kelp fraction cover (ground-truthing) is logistically challenging to obtain in remote locations or in the absence of high-resolution imagery, many classes might result in the model underestimating the predicted error. On the contrary, a small tree might not be able to classify new values [33]. Classification algorithms have thus proven effective but, due to their statistical data-adaptive nature, they are more reliable in local areas and less scalable to broader spatial scales, unless a large, spatially comprehensive (global) collection of training data [34]—not currently existing for this ecosystem—is available. For these reasons, there are still large areas where giant kelp forests remain undetected, and the lack of a baseline distribution map hampers the detection of their temporal trends. In the last global assessment of kelp forest change, only 34 of the 99 marine ecoregions with kelp were surveyed, and, of these, only nine had giant kelp data records longer than 5 years; none of them were from Sub-Antarctic ecoregions [10].

Key recent developments show the potential to revolutionise the remote detection of the global distribution of giant kelp. These are:

(a) New sensors onboard the recently launched European Space Agency's (ESA) Sentinel satellites offer an alternative to Landsat in the remote detection of giant kelp forests. Although individual kelp blades can show different photo-acclimation responses to variable conditions of light, the concentration of pigment chlorophylls *a* and *c* and fucoxanthin (Fuc) give the plants a conspicuous brownish colour, with a higher concentration with increasing depth [35]. These characteristics result in the highest spectral reflectance of kelp canopies being in the Near-Infrared (780–890nm) [19], the lowest being in the blue (400–500nm) and red (675nm) areas [35], with reflectance increasing strongly in the red-edge area [16]. The sensors onboard Sentinel-2, with four bands of 10 m and three additional red-edge bands of 20 m of spatial resolution, can strongly contribute to highlight this spectral area with a level of detail that other multispectral sensors do not possess, such as those in Landsat (30 m of spatial resolution).

(b) Cloud-based platforms such as Google Earth Engine (GEE, [36]) enable access to petabytes of open-access satellite imagery paired with an interactive development environment. Previous research in GEE in similar environments includes the analysis of the capabilities of Landsat imagery to detect kelp forests in British Columbia [17] and the use of Sentinel-2 imagery to estimate satellite-derived bathymetry [37] and to map seagrass [34]. Using GEE helps in processing large amounts of data to detect spatially persistent areas of giant kelp forests, taking into consideration that persistent kelp areas tend to occur at the centres of the forests, whereas borders are more variable [38]. This persistence has been associated with abiotic factors, such as water depth, the presence of a rocky substrate, substrate topology, and connectivity between the forests [38]. In contrast, variability is associated with ocean dynamics, such as wave height and sea-surface temperature ([14], more than seasonality), tidal ranges, or zenith angles [18]. Therefore, the central areas of forests should provide optimal material to build a global map of giant kelp.

(c) The use of Unmanned Aerial Vehicles (UAV) for coastal habitat mapping is a simple, cost-effective and reliable technology [39] that has been successfully used to map and validate intertidal biogenic reefs [40], saltmarsh biomass [41], and algal blooms [42]. Recent surveys to detect macroalgae in temperate coastlines have shown that RGB (additive primary colors—red, green, and blue—model) and multispectral cameras mounted on UAVs produce accurate imagery able to detect water turbidity and a range of taxonomical groups of algae in surface or shallow water, with the exception of spectrally similar species [18,43]. To our knowledge, there are yet no standardized protocols for marine or coastal mapping with UAVs [42].

The present study capitalises on these advances to produce an algorithm based on a series of filters to detect giant kelp forests at different latitudes and coastal contexts, to generate a global map of their distribution. We analysed the multi-band reflectance values of a set of classes or Regions of Interest (ROI) and defined a threshold-based approach that sequentially refined the kelp reflectance signal. This filtering algorithm was tested with three different spectral indices: the widely used Normalised

Difference Vegetation Index (NDVI [21], an index used in remote sensing to detect vegetation based on the differential reflectance of vegetation between wavelengths within the photosynthetically active radiation and the near infrared); the Floating Algae Index (FAI [44], an index specifically designed to detect floating algae); and the Kelp Difference (KD) we introduce in this paper. Spectral signatures were extracted from Sentinel 2 L1C imagery; masking thresholds were globally applied in Google Earth Engine, employing an average image composition from 2015 to 2019. Selected forests detected with this algorithm were validated at high-resolution with UAV-obtained orthomosaics. Our methodology successfully isolates giant kelp from the surrounding ocean and land except for green algae (*Ulvophyceae*). *Ulvophyceae* is a conspicuous class of algae found in intertidal environments such as estuaries or rocky areas [45] that has a similar spectral reflectance to giant kelp [43] and can share the same habitat only when giant kelp forests occur on shallow rocky intertidal areas [18]. Hence, in these geographical contexts, our map cannot distinguish between giant kelp and *Ulvophyceae*. As a result, a global map of giant kelp forests and intertidal algae is presented. The code is available in GEE, and the resulting map can be accessed online (https://biogeoscienceslaboxford.users.earthengine.app/view/kelpforests).

## 2. Materials and Methods

### 2.1. Training Data

We used Sentinel-2 Level 1C (Copernicus Service information) imagery provided by Google Earth Engine (GEE). Sentinel-2 L1C consists of ortho and radiometrically corrected images showing Top of Atmosphere (TOA) reflectance originally scaled by 10,000 [46] and rescaled by $10^{-4}$ for this study. Training areas consisted of previously studied giant kelp forest regions located at different latitudes and oceans, identified with their respective coordinates. An average $0.01° \times 0.01°$ image was produced for each selected area employing the tool developed by [47] to remove areas affected by cloud cover. This involved scanning the complete Sentinel-2 L1C dataset (23rd June 2015 to 31st July 2018) with cloud cover < 90%. The number of images processed thus varied for each site (Table 1). The final composite image represents 3 years of cloud-free grid cells. It emphasizes the temporal continuity of grid cell reflectance and includes all the possible zenith angles in this lapse of time. For each image, the following ROIs or classes were selected based on the literature and field experience: *Giant Kelp*, defined as giant kelp canopy; *Coast*, defined as grid cells laying between 0 and 1 m of elevation—such as rocks, beaches, human artefacts (e.g., docks, vessels), and any other elements close to the coastline and not permanently covered by water; *Ocean*, defined as areas of open ocean surface; *Foam*, defined as grid cells in an extensive (10 km) surf zone; *Organic water*, defined as highly productive waters; *River grass* in an estuarial area; *Green algae*, i.e., intertidal green algae (*Ulvophyceae*); and *Land vegetation* that can be found between 0 and 1 m above sea level and could be misinterpreted as kelp grid cells. We sampled 90 points per ROI, totalling a training data set of 720 observations (Table 1).

**Table 1.** Sites selected as training data in this study.

| Class (ROI) | Site | Long, Lat | Images Processed | Reference |
|---|---|---|---|---|
| | S. Australia—Warrnambool | 142.46, −38.4 | 163 | [48] |
| | S. Africa—Oudekraal | 18.35, −33.98 | 310 | [48] |
| | Falkland Islands (Malvinas) | −57.75, −51.61 | 73 | [48] |
| | W. Canada—Nuchatlitz Islands | −126.53, 49.6 | 258 | [48] |
| Kelp, Coast, Ocean | USA—Carmel Bay | −121.93, 36.55 | 76 | [48] |
| | C. Chile—Punta Parra | −72.97, −36.66 | 171 | [49] |
| | S. Chile—Grevy Island– Cape Horn | −67.61, −55.52 | 93 | [29] |
| | France—Kerguelen Islands | 69.68, −49.20 | 98 | [50] |
| | South Georgia & the South Sandwich Islands | −36.71, −54.11 | 129 | [51] |
| Foam | S. Chile—Queule | −73.21, −39.35 | 77 | This study |
| River Grass | S. Chile—Queule | −73.21, −39.35 | 77 | This study |

**Table 1.** *Cont.*

| Class (ROI) | Site | Long, Lat | Images Processed | Reference |
|---|---|---|---|---|
| Land vegetation | S. Chile—Queule | −73.21, −39.35 | 77 | This study |
| Green algae (*Ulvophyceae*) | S. Chile—Puyuhuapi Channel | −72.76, −44.71 | 78 | This study |
| | S. Argentina – Puerto Deseado | −65.86, −47.85 | 147 | |
| Organic water | USA—Santa Barbara Channel | −119.95, 34.03 | 282 | This study |
| | S. Chile—Puyuhuapi Channel | −72.71, −44.73 | 78 | |
| | New Zealand—Kaimaumau | 173.31, −34.96 | 136 | |

For each ROI, the multispectral reflectance of their respective grid cells was extracted from the averaged image using R (packages: raster, rgeos, sp, maptools, dplyr), and mean and standard deviation were calculated per class for all Sentinel-2 Bands (Figure 1).

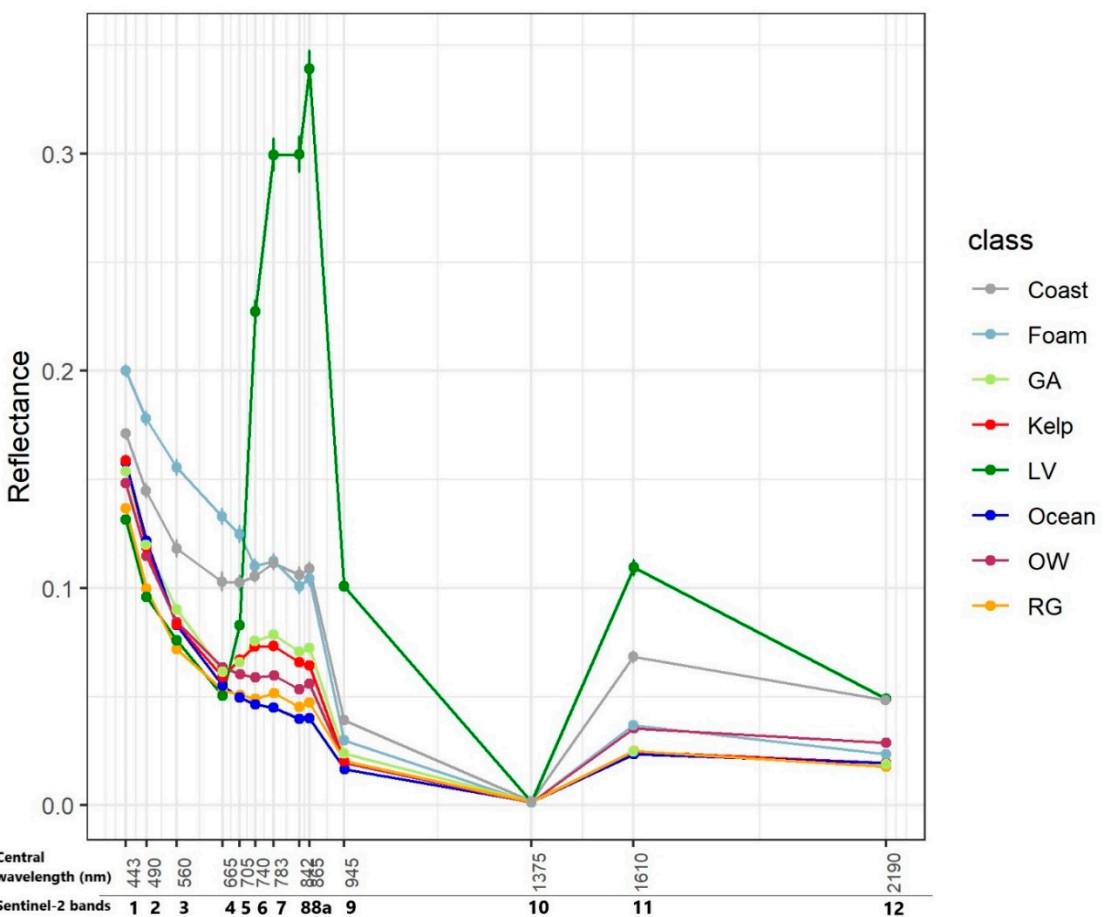

**Figure 1.** Mean ± SD reflectance values for each Region of Interest (ROI) used as training data in this study, computed for all Sentinel-2 bands (central wavelength of each band (in nm) and band number indicated in the × axis). Classes or ROIs: *Coast*; *Foam*; GA = *Green Algae*; *Kelp*; LV = *Land vegetation*; *Ocean*; OW = *Organic water*; RG: *River grass*.

### 2.2. Kelp Filter Algorithm

A 3-step process was followed to optimise giant kelp identification from the multispectral reflectance values of the ROIs:

1. **Band-based threshold.** The multispectral profiles of Land Vegetation, Coast, and Foam are clearly distinctive (Figure 1). One hundred percent of all Coast and Land Vegetation observations in the training data were larger than or equal to B11 = 0.028, which corresponds to the minimum value of Coast ROI at B11 (1610 nm at central wavelength, Figure 2). Consequently, all observations

with value B11 ≥ 0.028 were masked out, which resulted in 305 training data grid cells remaining. This eliminated 21 observations in the upper quartile of the original kelp sample (Figure 2). This was done to avoid misclassification with coastal features at the expense of identifying some kelp-occupied grid cells with higher-than average reflectance values in band B11. These were found to be marginal portions of the identified giant kelp forest ROIs—i.e., grid cells occupying the periphery of kelp stands.

2. Kelp Difference (KD). Giant kelp grid cells exhibited a conspicuous large difference in reflectance between bands in the red edge area of the spectrum (Bands 5, 6 and 7) and the red band (B4). Selecting B6 (central wavelength = 740 nm) as the band with the largest difference with B4 (Figure 1), we defined a Kelp Difference (KD) as the difference between both band values. *Step 2* was applied after the band-based masking (*Step 1*), although the order of these two steps would not alter the result:

$$KD = (R_{B6} - R_{B4}) \qquad (1)$$

3. KD-based threshold. A second masking threshold was applied to the KD-converted training dataset. This enabled 100% of the grid cells not belonging to *Giant Kelp* or *Green Algae* to be removed (Table 2 and Appendix A). The reflectance values for *Giant Kelp* and *Green Algae* were found to be too similar to be efficiently discriminated. In order to compare the performance of the KD in relation to other indices used to remotely detect algae in the past, we separately implemented this step employing NDVI and FAI. This resulted in the production of three different kelp maps.

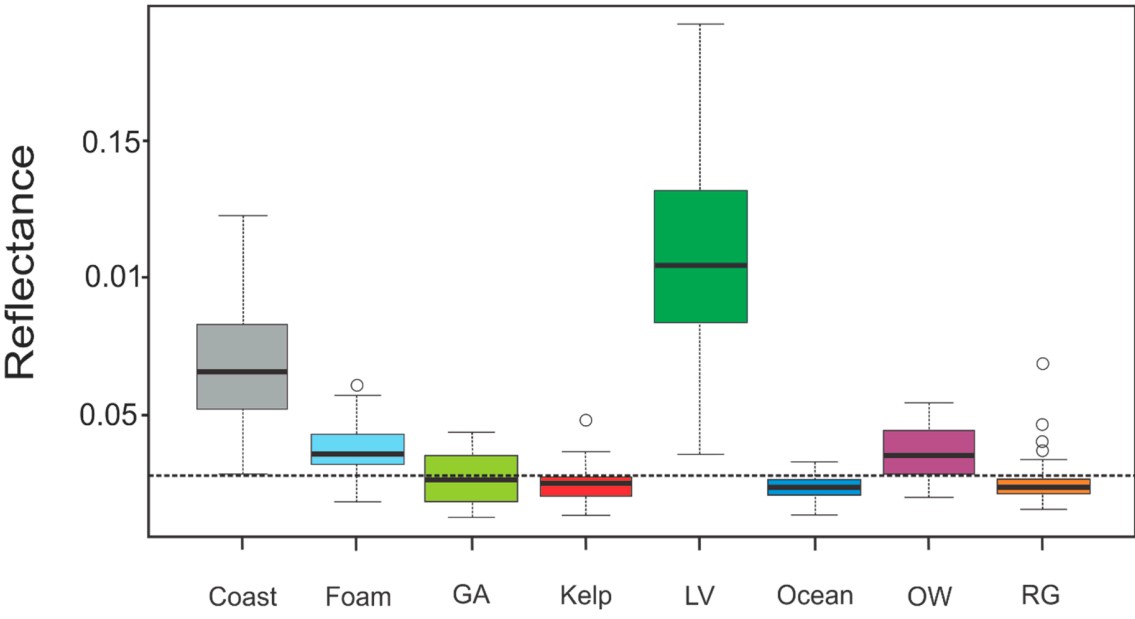

**Figure 2.** Box and whisker plot of B11 reflectance values (90 grid cells per class). All grid cells with B11 ≥ 0.028 (indicated with a dashed line) were masked out to separate *Coast* and *Land Vegetation* from the other ROIs. Classes or ROIs: *Coast*; *Foam*; GA = *Green Algae*; *Kelp*; LV = *Land vegetation*; *Ocean*; OW = *Organic water*; RG: *River grass*.

**Table 2.** Masking out threshold by index.

| Index | Masking Threshold | Defined by |
|-------|-------------------|------------|
| NDVI | ≥−0.003411 | *River grass* max value |
| FAI | ≥0.005352 | *Organic water* max value |
| KD | ≥0.003216 | *River grass* max value |

*2.3. Validation at High Spatial Resolution*

　　Independent data of giant kelp forests were collected along the Chilean coastline and the Falkland Islands (Malvinas) covering a latitudinal distance of 2470 km. This distance accounts for the wide range of environments where giant kelp occurs in different climates—from warm temperate (Csb, Cfb and Cfc) to tundra (ET) according to the Köppen-Geiger climate classification [52]—and from the Pacific to the Atlantic oceans (Table 3, Figure 3). The ground-truthing process was conducted during the austral summer and winter of 2019, between 24th January and 18th February in the Strait of Magellan, 21st February to 9th March in the Patagonian Channels and Fjords, 15th March in Niebla, 24th March in Maitencillo, 11th July in the Tussac Islands, and 21st July in Yendegaia. For each site, a mapping mission flew a DJI-Phantom 4 pro unmanned aerial vehicle (UAV) with an RGB camera mounted on it (FOV 84° 8.8 mm/24 mm (35 mm format equivalent) f/2.8–f/11 auto focus at 1 m–∞). The overlap of the images was 80%, and the height was between 70 and 100 m depending on coastal topography. The time of the day for the flights was between 10 a.m. and 4 p.m. The aerial images were processed using either the application DroneDeploy or Agisoft Metashape (in the case of the Falkland Islands). Any potential drone orthomosaic error was visually assessed in GEE to be of minimal impact on validation comparisons. Because of bad weather conditions during the fieldwork campaign in the Strait of Magellan, the giant kelp forests of Buque Quemado, San Gregorio, and Chabunco were only partially mapped. In each ground-truth orthomosaic (10 cm spatial resolution), kelp forests—defined as the presence of clearly visible canopies in the RGB high-resolution image—were visually identified and delineated in the form of polygon shapefiles. They were then converted into a binary raster format: kelp and no-kelp, at 10 m of spatial resolution. Ground-truthed layers were compared with the filtered kelp grid cells calculated separately for NDVI, FAI, and KD over the same extents at 10 m of spatial resolution, averaged over the 3 months prior to the survey and converted to binary values of kelp and no-kelp using the values in Table 2 as thresholds.

**Table 3.** Summary of validation sites, their coordinates, the extent of the kelp stand (ha), and climate zone. Climate acronyms: Csb: Warm temperate climate with dry and warm summer. Cfb: Warm temperate climate, fully humid and warm summer. Cfc: Warm temperate climate, fully humid, and cold summer. ET: Polar tundra [52].

| Main Area | Site | Long, Lat | Area (ha) | Köppen−Geiger Climate Classification |
|---|---|---|---|---|
| C. Chile | Maitencillo | −71.44199, −32.64762 | 0.5 | Csb |
| S. C. Chile | Niebla | −73.40054, −39.87498 | 2.1 | Cfb |
| Channels and Fjords | Lobera María Isabel | −73.42381, −44.90923 | 0.3 | Cfb |
| Channels and Fjords | San Andrés 1 and 2 | −73.32865, −44.9348 | 0.3 1 | Cfb |
| Channels and Fjords | Puerto Amparo | −73.28257, −44.89874 | 0.3 | Cfb |
| Strait of Magellan | San Isidro | −70.97483, −53.78515 | 5.0 | Cfc |
| Strait of Magellan | Santa Ana Sur | −70.92467, −53.63006 | 0.4 | Cfc |
| Strait of Magellan | Santa Ana Norte | −70.91918, −53.62731 | 1.1 | Cfc |
| Strait of Magellan | Punta Carrera | −70.93902, −53.55859 | 5.2 | Cfc |
| Strait of Magellan | Chabunco | −70.81101, −52.98648 | 1.4 | Cfc |
| Strait of Magellan | San Gregorio | −70.07255, −52.57044 | 0.8 | Cfc |

**Table 3.** *Cont.*

| Main Area | Site | Long, Lat | Area (ha) | Köppen−Geiger Climate Classification |
|---|---|---|---|---|
| Strait of Magellan | Buque Quemado | −69.47702, −52.33489 | 1.2 | Cfc |
| Beagle Channel | Yendegaia | −68.70262, −54.9045 | 1.5 | ET |
| Falkland Islands (Malvinas) | Tussac Islands: Kelly Rocks, Bottom, Top | −57.74472, −51.67233 | 11.5 16.3 11.8 | ET |

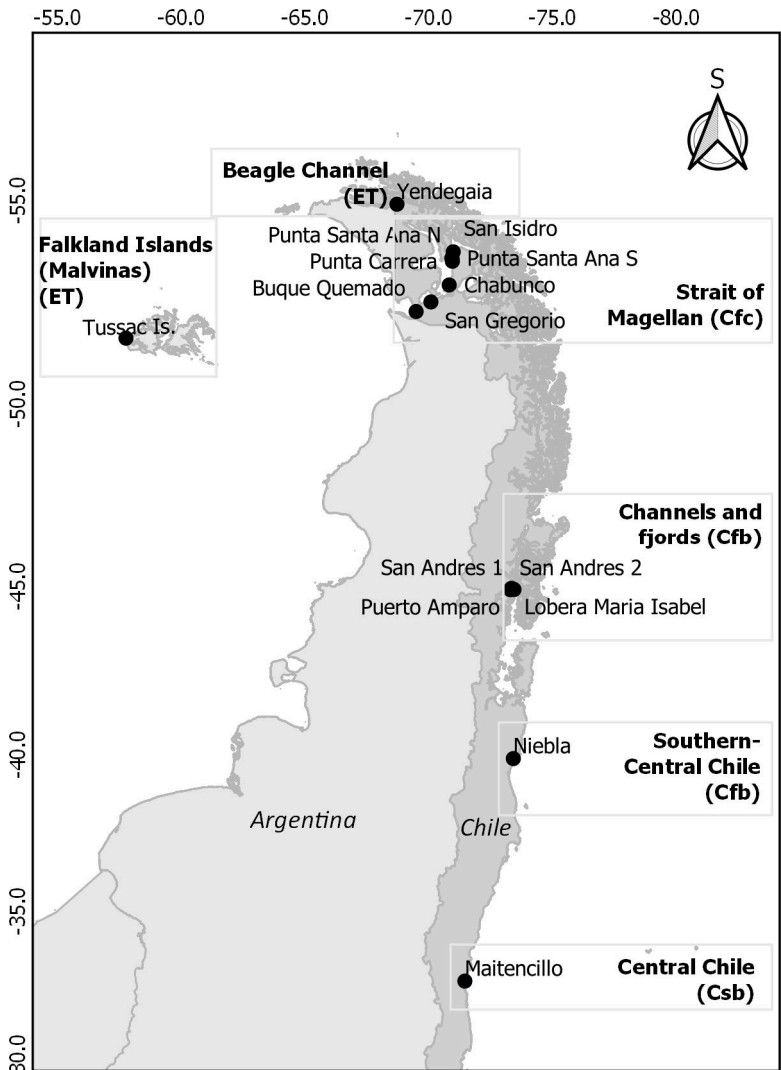

**Figure 3.** Map of the surveyed sites for validation at a high resolution of giant kelp forests. Cartographic projection EPSG: 3857. Maps in this article are south-oriented to highlight the connectivity of giant kelp forests at Sub-Antarctic latitudes.

Performance statistics based on confusion matrices (one matrix per pair of ground-truthed data/remote sensing kelp maps, i.e., three in total) were computed in ENVI 4.7. These were Producer's, User's, and Overall Accuracy plus the Cohen's kappa [53] statistics.

*2.4. Kelp Filter Algorithm in Google Earth Engine*

The kelp detection product with the best overall accuracy and kappa coefficient was applied over the temperate coastlines of the planet using GEE through the following steps:

(a) cloud-free tool of [47] over Sentinel-2 grid cells scaled at $10^{-4}$ from 26th June 2015 to 23rd June 2019;

(b) kelp filter threshold;

(c) masking of all grid cells with elevation above sea level > 0 m using two digital elevation models: Advanced Land Observing Satellite (ALOS) and Shuttle Radar Topography Mission (SRTM), both at 30 m of spatial resolution. This last procedure was done to avoid any misclassification of elements on land with a similar reflectance to giant kelp that were not included in our ROI training data set. To improve the readability of the index, digital numbers were rescaled to values from 0 to 255 in the maps.

The ALOS digital elevation model fills the void grid cells of other digital elevation models and thus has higher accuracy than SRTM (more information in [54]). ALOS in GEE does not cover the following archipelagos: Chatham, Prince Edward, Tristan da Cunha, Gough Island, and the extreme NW of South Georgia. In these cases, SRTM was used. No digital elevation models were available for the Diego Ramirez archipelago. In this case, a polygon shapefile of the coastline was used as a mask. As Sentinel-2 imagery does not cover some of the Southern Ocean islands (Amsterdam and Saint-Paul, Bouvet, Antipodes, and Bounty Islands), those areas had to be excluded from the map. The code for the GEE algorithm in JavaScript is available in Appendix A.

### 2.5. Validation at a Low Spatial Resolution

To validate the global map, we measured the linear distance between kelp grid cells and previously surveyed or observed giant kelp forests. To this end, we employed a dataset of 157 locations in South and North America, New Zealand, Southern Australia and Tasmania, and South Georgia and the sub-Antarctic islands, recorded in a multipoint layer in Google Earth (Figure 4 and Appendix A). As these stands were identified at different times and degrees of accuracy, the linear distance in breaks of 50, 100, 200, and 300 m was measured from each point to the closest mapped kelp grid cell. A workflow of this methodology is shown in Appendix A. The detected grid cells were summarized by marine ecoregion and province following the classification of [55].

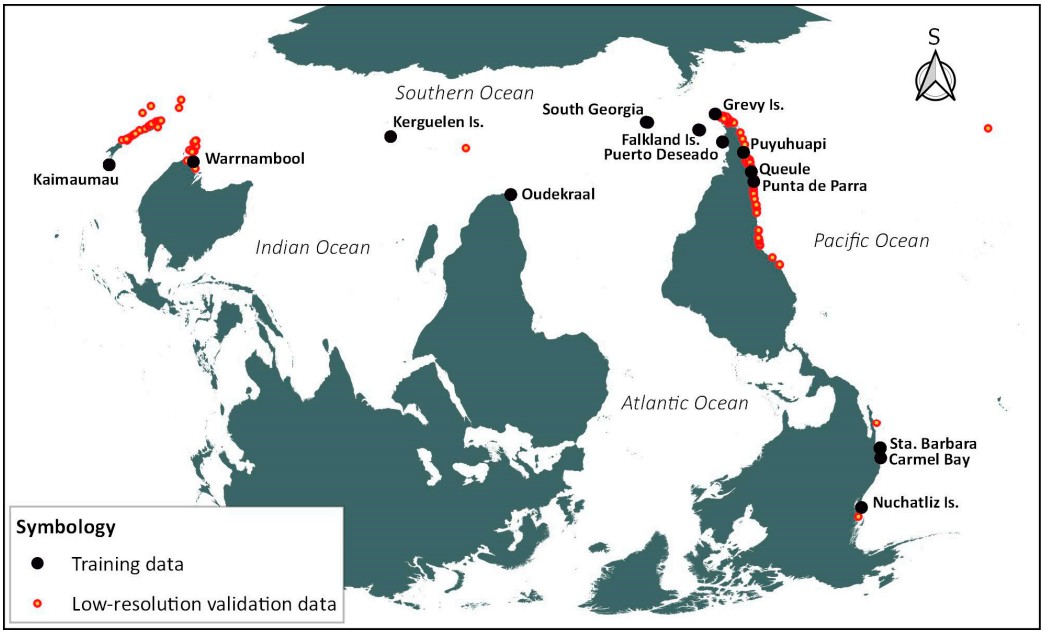

**Figure 4.** Location of the training data (black dots) and the low-resolution validation sites (red dots). Cartographic projection EPSG: 54042. Maps in this article are south-oriented to highlight the connectivity of giant kelp forests at sub-Antarctic latitudes.

## 3. Results

### 3.1. Kelp Filter Algorithm

Using the original training data of 90 grid cells per ROI (720 grid cells in total), the applied protocol preserved a fraction of *Giant kelp* grid cells, which were concentrated at the centre of the observed kelp canopies and thus represented the purest grid cells or at least the grid cells most dominated by kelp canopy, and *Green algae* grid cells on rocky intertidal areas (Table 4). Excluded *Giant kelp* grid cells were located at the peripheries of identified *Giant kelp* stands, where they were more likely to display mixed reflectances with *Ocean*, *Foam*, *Organic water*, or *River grass* in some areas. Zero grid cells belonging to any of the other ROIs remained after the application of the kelp filter algorithm. The use of the KD in the algorithm preserved the highest quantity of *Giant kelp* grid cells in the training area, followed by FAI and NDVI. As our algorithm cannot distinguish between *Giant kelp* (*M. pyrifera*) and *Green algae* (*Ulvophyceae*), our resulting map shows *M. pyrifera* and rocky intertidal *Ulvophyceae* cover.

**Table 4.** Remaining grid cells out of 90 *Giant kelp* and 90 *Green algae* original observations (total = 180) after the application of the kelp filter algorithm to the training data.

| Index | No. of Observations Total | No. of Observations Kelp | No. of Observations Green Algae |
|-------|---------------------------|--------------------------|--------------------------------|
| NDVI  | 102                       | 50                       | 52                             |
| FAI   | 103                       | 50                       | 53                             |
| KD    | 114                       | 61                       | 53                             |

### 3.2. Validation at High Resolution

The ground-truthing process is illustrated in Figure 5, where filtered grid cells are overlain on a UAV orthomosaic that is used as ground-truthed imagery (see Appendix A for the complete collection of UAV imagery). Maps obtained employing KD, FAI, and NDVI show higher values at the centre of the giant kelp canopy, with decreasing values towards their periphery. The levels of sensitivity are different with each index, with NDVI showing more saturation and homogeneous values over the canopy.

The confusion matrix results (Table 5) show similar levels of accuracy for all forests > 1 ha, with an overall accuracy (OA) ≥ 78% and a kappa coefficient ≥ 0.52 for the three indices, and with a slightly better overall kappa coefficient for maps obtained using KD (mean = 0.66) over FAI and NDVI (means of 0.65 and 0.64, respectively). Smaller kelp forests (Maitencillo, Lobera María Isabel, San Andrés 1 and 2, Santa Ana Sur) showed only sparse grid cells over the ground-truthed areas, and their levels of accuracy were much lower. The forest of Puerto Amparo, which is composed of a mostly underwater canopy, was only detected as sparse grid cells over the area. Finally, the forests of San Gregorio and Buque Quemado, located in areas of high tidal range (9 m) were undetected in the averaged image.

Regarding the Producer's and User's accuracy metrics, the three maps show percentages of accuracy > 51% for all the sites > 1 ha, with averages ≥ 64% for PA and ≥ 81% for UA (Table 5). In agreement with the conservative approach used in our kelp filter algorithm (i.e., our algorithm aimed at minimising false detections), the Producer's accuracy (an indicator of omission error or underestimation of the canopy extent) is slightly lower than the User's accuracy (that indicates commission errors or the overestimation of the canopy on the water surface). The map with the least underestimation was obtained using the KD (73.1% Producer's Accuracy) and the map with the least overestimation was obtained using FAI (82.8% User's Accuracy). The KD showed the highest average and the most balanced (lowest standard deviation) combination of accuracy metrics.

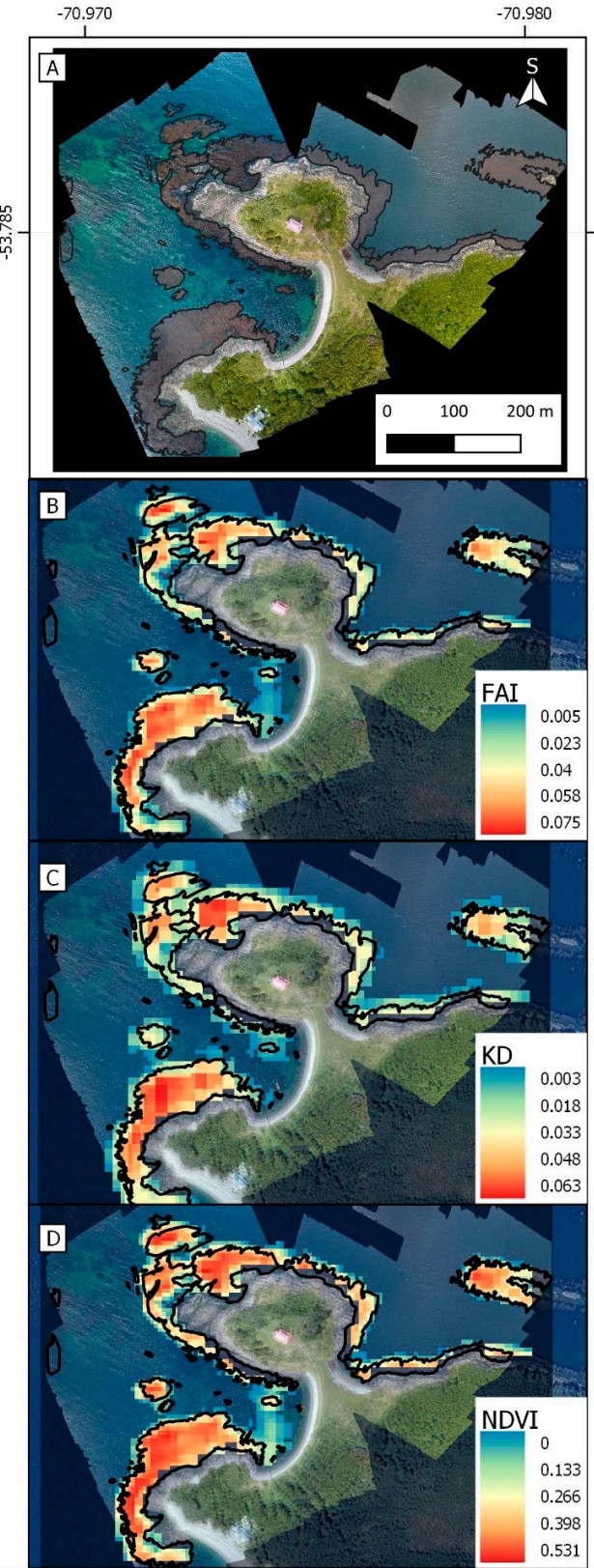

**Figure 5.** Ground-truthing process. A: 'True' canopy is delineated as the black line in the UAV orthomosaic. B, C, and D: maps obtained through the kelp filter algorithm applied on the Floating Algae Index (FAI), the Kelp Difference (KD) and the Normalised Difference Vegetation Index (NDVI), respectively. Higher values show higher concentrations of kelp canopy, although each index shows different levels of sensitivity. Site: San Isidro (Strait of Magellan). Cartographic projection EPSG: 3857, South-oriented.

**Table 5.** Confusion matrix results for maps obtained using the kelp filter algorithm employing the FAI, NDVI, and the KD in forests larger than 1 hectare. PA = Producer's Accuracy (%). UA = User's Accuracy (%). OA = Overall Accuracy (%). Overall Accuracies > 80 and Kappa coefficients > 0.6 are highlighted in bold.

| | FAI | | | | NDVI | | | | KD | | | |
|---|---|---|---|---|---|---|---|---|---|---|---|---|
| | **PA** | **UA** | **OA** | **Kappa** | **PA** | **UA** | **OA** | **Kappa** | **PA** | **UA** | **OA** | **Kappa** |
| **Punta Santa Ana Norte** | 59.3 | 84.3 | **92.3** | **0.65** | 61.2 | 80.0 | **91.9** | **0.65** | 62.5 | 75.0 | **91.3** | **0.63** |
| **San Isidro** | 69.2 | 76.7 | **92.9** | **0.69** | 71.0 | 74.4 | **92.7** | **0.68** | 75.1 | 67.4 | **91.6** | **0.66** |
| **Chabunco** | 69.2 | 76.8 | **92.7** | **0.69** | 72.0 | 73.9 | **92.5** | **0.69** | 82.7 | 64.5 | **91.2** | **0.67** |
| **Punta Carrera** | 76.0 | 94.6 | **91.9** | **0.79** | 78.7 | 93.7 | **92.4** | **0.80** | 84.2 | 92.9 | **93.7** | **0.84** |
| **Niebla** | 62.6 | 79.7 | **80.9** | 0.56 | 66.5 | 77.0 | **80.9** | 0.57 | 67.9 | 69.8 | 78.0 | 0.52 |
| **Tussac Kelly** | 66.2 | 89.5 | **86.2** | **0.67** | 61.0 | 90.9 | **85.0** | **0.63** | 72.8 | 87.4 | **87.5** | **0.71** |
| **Tussac Bottom** | 60.8 | 88.4 | **89.4** | **0.66** | 57.6 | 88.4 | **88.8** | **0.63** | 71.4 | 85.5 | **90.8** | **0.72** |
| **Tussac Top** | 69.1 | 74.6 | **80.8** | 0.57 | 60.8 | 76.4 | 79.6 | 0.53 | 77.6 | 73.6 | **82.3** | **0.62** |
| **Yendegaia** | 56.1 | 80.8 | **89.3** | **0.60** | 51.2 | 82.1 | **88.8** | 0.57 | 63.2 | 71.2 | **88.4** | **0.60** |
| **Total average** | 65.4 | 82.8 | **88.5** | **0.65** | 64.4 | 81.9 | **88.1** | **0.64** | 73.1 | 76.4 | **88.3** | **0.66** |

### 3.3. Validation at Low Resolution

We computed the global map employing the KD, since it produced the best validation statistics. For a total of 157 georeferenced sites, 59 (37.6%) were found to be within a range of 100 m, 103 (65.6%) within a range of 300 m, and 54 sites (34.4%) more than 300 m away from a reported kelp stand (Table 6). The ecoregions with the lowest detection (≤50%) were Humboldtian, Western Bassian, Bounty-Antipodes, and Central New Zealand, whereas all other ecoregions had ≥ 66% of detection. Considering that this is a global map at a grid cell resolution of 10 m, our results indicate a moderate to strong ability of this filtering algorithm to detect areas with kelp within a range of 300 m, with more success at higher latitudes and larger kelp forest areas. Table 7 provides a summary of the detected area of giant kelp and intertidal green algae per marine ecoregion and province.

**Table 6.** Summary of the range distance (m) between observed sites and the closest kelp grid cell.

| Range (m) | N | % | Cumulative% |
|---|---|---|---|
| 50 | 42 | 26.8 | 26.8 |
| 100 | 17 | 10.8 | 37.6 |
| 200 | 27 | 17.2 | 54.8 |
| 300 | 17 | 10.8 | 65.6 |
| >300, undetected | 54 | 34.4 | 100.0 |
| Total | 157 | 100 | |

**Table 7.** Area in km$^2$ per marine ecoregion and province with detected giant kelp and intertidal green algae, and the number of detected grid cells versus the total of the georeferenced sites per ecoregion in a range of 300 m.

| Province | Ecoregion | km$^2$ | Detected/Georeferenced |
|---|---|---|---|
| Agulhas | Agulhas Bank | 136.2 | |
| | Natal | 1.7 | |
| Benguela | Namaqua | 96.5 | 1/1 |
| Cold Temperate Northeast Pacific | Gulf of Alaska | 483.9 | |
| | North American Pacific Fjordland | 2074.2 | |
| | Northern California | 193.7 | 1/1 |
| | Oregon, Washington, Vancouver Coast and Shelf | 333.6 | |
| | Puget Trough/Georgia Basin | 118.2 | |
| Magellanic | Channels and Fjords of Southern Chile | 4840.7 | 23/32 |
| | Chiloense | 687.0 | 17/22 |
| | Falkland Islands (Malvinas) | 3081.1 | 1/1 |
| | Patagonian Shelf | 144.5 | |

**Table 7.** *Cont.*

| Province | Ecoregion | km$^2$ | Detected/Georeferenced |
|---|---|---|---|
| Northern New Zealand | Northeastern New Zealand | 76.6 | |
| | Three Kings–North Cape | 0.6 | |
| Scotia Sea | South Georgia & the South Sandwich Islands | 145.9 | 2/2 |
| Southeast Australian Shelf | Bassian | 389.3 | 10/13 |
| | Cape Howe | 128.4 | |
| | Western Bassian | 42.4 | 1/3 |
| Southern New Zealand | Central New Zealand | 75.7 | 5/11 |
| | Chatham Island | 23.3 | 1/1 |
| | Snares Island | ND | |
| | South New Zealand | 148.8 | 7/8 |
| Subantarctic Islands | Crozet Islands | 73.6 | |
| | Heard and McDonald Islands | 0.5 | |
| | Kerguelen Islands | 3397.6 | |
| | Macquarie Island | 17.2 | |
| | Prince Edward Islands | 46.4 | 1/1 |
| Subantarctic New Zealand | Auckland Island | 29.1 | 1/1 |
| | Bounty and Antipodes Islands | 0.6 | 1/2 |
| | Campbell Island | 1.5 | 1/1 |
| Tristan Gough | Tristan Gough | 6.0 | |
| | Southern California Bight | 222.1 | |
| Warm Temperate Southeastern Pacific | Araucanian | 54.9 | 12/18 |
| | Central Chile | 11.7 | 9/10 |
| | Humboldtian | 4.7 | 9/29 |

## 4. Discussion

### 4.1. Kelp Detection and Mapping

Our methodology simplifies the giant kelp detection process by refining the parameters to detect canopies. Although we recommend its use with the KD, the method can be done using well-known and widely used indices such as the NDVI and FAI, which can be applied to databases with longer records than Sentinel-2, such as Landsat, with both averaged and non-averaged imagery. However, this algorithm has limitations: seven out of 17 high-resolution and 54 out of 157 low-resolution validation sites used in this research were not detected. At a high resolution, the best results occurred when detecting giant kelp forests ≥ 1 hectare with Sentinel-2 imagery at 10 m spatial resolution. This represents a large improvement in relation to the Landsat 30 m resolution: in the Santa Barbara Channel where [56] found a mean patch size of 0.28 km$^2$ (28 ha), a much larger value than the largest observed forest in the Falklands training data in this study (Bottom Island, 16.3 ha). A similar minimal mapping unit of 1 hectare was recommended for the Global Mangrove Watch initiative, made with a fusion of multispectral (Landsat) and ALOS imagery classified with the Extremely Randomized Trees algorithm. In this case, more than 50,000 training points were used [57,58]. As would be expected due to the limited training data, applying our training data to classification algorithms like CART or Random Forests yielded poor results and hence were not included in the main results of this study (they can be examined in Appendix A). The sources of *River grass*, *Green algae* and *Organic water* observations in our training dataset were limited to specific areas because of our local knowledge and sources of secondary data. The training dataset volume used in this study is in line with previous remote sensing studies that have been carried with similar training numbers, e.g., 17 sampling sites [59] and five sampling classes [60], both with high levels of accuracy when tested with independent field data. Like ours, these studies avoided the use of machine-learning algorithms, since they work well with large amounts of training data but show lower performance otherwise, and proposed instead working with combinations of reflectance bands as an optimal way to detect targeted ROIs. Further research is

required to establish which method is best with future increases in ground data availability, as well as to detect other coastal and marine ecosystems (e.g., subtidal kelp forests).

The low-resolution validation represents a very demanding test for the kelp filter algorithm, and its results need to be placed in perspective. First of all, the spatial precision of the validation data points is variable, and hence observed accuracies between the mapped distribution and the published kelp canopies of <300 m are notable. Second, the array of low-resolution stands includes observations made over a period of 15 years, with the oldest ones dating from 2004 and 75% being made before the start of the satellite observation period used in the current study. Third, the metadata in the low-resolution calibration test does not contain information on the size of the reported kelp stands, which can be smaller than 1 ha and are thus not detectable with this method. Lastly, our 4-yr averaged map does not detect seasonality in the canopy extent and shows only permanent grid cells over the same area. Highly variable canopies may thus be undetected. The finding that 65.6% of the reported low-resolution sites fall within 300 m of KD-detected stands—which are based on spatial resolutions of 20 m (B6) and 10 m (B4)—needs to be interpreted under this light.

### 4.2. Spatial and Biophysical Patterns

Maps produced with the kelp filter algorithm show larger values at the centre of the forests and lower values at their peripheries in the validation sites, as reported for the giant kelp forests characterized by [38] and [13]. This may indicate mixed spectral signatures on the edges of the kelp stands due to lower frond density, as it was observed in fieldwork and with our high-resolution UAV images. Edge effects in kelp forests have an important impact on ecological processes (e.g., herbivory) as well as on mechanical action (e.g., turbulence) in and around the forest [7].

Previous research has identified low average NDVI values in kelp canopy forests (<0.30)—[13] and [17] used a threshold of NDVI > 0.05 in two or more scenes to detect kelp. Both results agree with the values found in our study sites. Considering the similarities of the maps created using the FAI, KD and NDVI, it can be hypothesized that the KD and FAI are related to biomass in a similar way to the NDVI, although more data are needed to test if the use of this algorithm would help to detect this biophysical variable. Previous research completed in the Santa Barbara Channel found correlations between giant kelp estimates of canopy biomass computed using Landsat imagery and adult plant density ($R^2 = 0.85$, $P < 0.001$; [61]) and total forest biomass ($R^2 = 0.73$, $P < 0.001$; [13]), and between the kelp fraction index and canopy biomass ($R^2 = 0.64$, $P < 0.001$; [14]), with total biomass values that ranged from $4.14 \times 10^6$ to $4.74 \times 10^8$ kg in 25 years (mostly affected by large winter storms) for a study area *c.* 1500 km in length [62]. The positive results obtained with Landsat hold promise for the use of Sentinel-2 imagery in a similar fashion in the near future.

On the other hand, the phenotypic plasticity of giant kelp implies that it can adopt different shapes and sizes in the ecomorphs '*pyrifera*' '*integrifolia*', '*angustifolia*', and '*laevis*', named after their differences in holdfasts and blade texture [9,11,48]. Although our global map seems to detect canopies in their complete distribution, the low number of detected sites in the Humboldtian ecoregion, where the ecomorph '*integrifolia*' occurs [9], requires more sampling sites to test our detection thresholds.

### 4.3. False Negatives

An important characteristic of giant kelp plants is that they form canopies on the surface, but this is not a mandatory rule. In fact, the plants of Buque Quemado in the Atlantic side of the Strait of Magellan are subjected to tidal ranges > 9 m and at high water tend not to be visible in Sentinel-2 imagery, while at low tide the plants are completely exposed (Figure 6A,B). In British Columbia, a 2 m increase in tide decreased the detected kelp extent by 40% [17], an effect that may also be reflected in our results (e.g., the intertidal forest of Niebla). However, tidal ranges < 4 m have shown not to affect or to produce large biases in the estimates of kelp forest extent, as observed by [13,14] in the Santa Barbara Channel forests. Coastlines with tidal ranges > 9 m, where giant kelp might be present are

restricted to the south-eastern coast of South America and western North America (British Columbia and Alaska) [63], and hence those areas could be underestimated in our global kelp map.

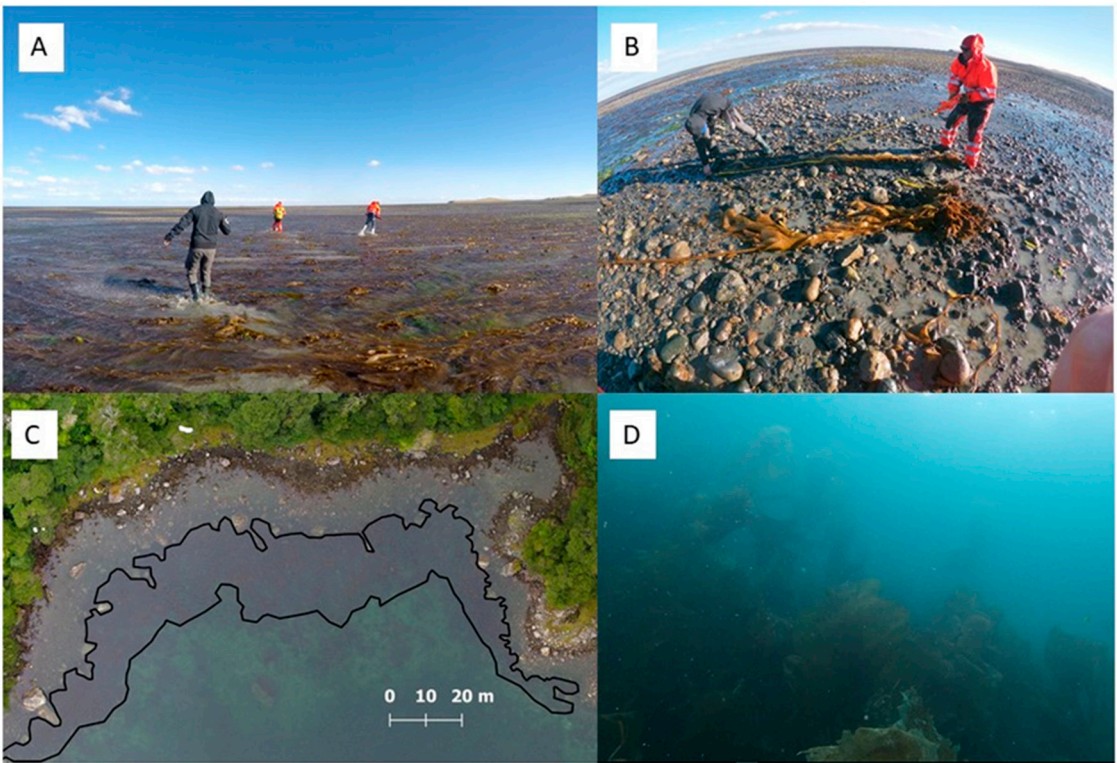

**Figure 6.** (**A**) Buque Quemado intertidal forest (Strait of Magellan) at 10 cm of depth in low tide and 9 m in high tide. (**B**) Measuring the length of the plants of site A. (The people in photos A and B are included for scale.) (**C**) A permanent subtidal forest (delineated in black) in Puerto Amparo (area = 0.3 ha). (**D**) The underwater canopies of Puerto Amparo (max. depth: 7.9 m). (Photos: Alejandra Mora S.).

Forests with few plants reaching the surface were not successfully detected, as was the case with Puerto Amparo in the Channels and Fjords area. In our in-situ observation, the plants looked stressed and not reproductive under the water (Figure 6C,D). There are other areas in the world where the canopy never reaches the surface, such as around the Channel Islands in Southern California [64]. A similar case was found in our study site of Maitencillo in central Chile, which could be explained by summer temperatures reaching sub-lethal levels with a monthly sea surface temperature between 17–18°C (OBPG, 2015) before our survey. According to [65], the mortality of the macroscopic sporophytes of giant kelp increases with temperatures > 15–17°C, following previously observed patterns of seasonality [9]. The low number of detected sites in the Humboldtian and Western Bassian ecoregion may be explained by kelp harvesting [66] or climate-driven shifts [67]. Finally, San Andrés 1 and Lobera María Isabel were small patches almost completely undetected by the spatial resolution of the Sentinel sensor.

*4.4. False Positives*

The presence of kelp canopies > 25 m$^2$ can be detected in a grid cell with 20 m spatial resolution [19], but the elements that fall into that instantaneous field of vision may cause commission errors. In this study, reflectance thresholds, a combination of values of key reflectance bands, and topography filters were used to isolate giant kelp grid cells, and our training data were focused on the presence of giant kelp, which is the only canopy-forming kelp species in South America, and *Ulvophyceae*, which is a conspicuous class of algae in rocky intertidal areas. This procedure did not mask out intertidal areas because giant kelp can also grow in these environments [9], therefore *Ulvophyceae* can

be considered as part of the positive error of this research, but restricted to rocky intertidal areas only. Due to lack of georeferenced data to generate ROIs for our calibration dataset, we did not include the intertidal-subtidal *Durvillaea antarctica*, a southern member of order Fucales that can sometimes reach the surface, or *Nereocystis luetkeana* of North America, which also forms canopies [15]. The same applied to the presence of floating (detached) specimens of *D. antarctica* and giant kelp. Floating kelp can reach > 1000 kg km$^2$ in the Patagonian Fjords [68]; these kelp rafts are not uniform, and temporal variation occurs [69]. Hence, it was assumed that their ephemeral occurrence was not detected in our 4-year composite images. The use of other sensors (most notably hyperspectral [70]) may help to separate actual giant kelp from the reflectances of other algae, or, alternatively, more ROIs in Sentinel-2 can help to confirm the capacity of this method to effectively detect intertidal algae.

### 4.5. Notes on UAV Surveying

Our high-resolution validation confirmed that RGB cameras mounted on UAVs can be successfully used for giant kelp monitoring [18]. The length of the canopy blades floating on water (minimum length of an individual blade = 13.85 cm, M. Palacios personal communication) makes fronds and canopies easily detectable with spatial resolutions of 10 cm. Integrating UAV orthomosaics and GEE-processed images in either a GEE or a Geographic Information System environment is a smooth process that can be easily used for ground-truthing satellite imagery of land or coastal classes. Some technical challenges observed in the field were that RGB cameras can be easily blinded by solar reflection on water. The use of a polarizing filter on the camera lens to cut solar flaring could be explored to avoid this problem. We also advise including a fraction of land on the surveyed area to facilitate the ground-control of the orthomosaic.

### 4.6. Additional Considerations about the Global Map

The global KD map was created using satellite imagery composited over four years. This represents the first complete high-resolution map of giant kelp, the distribution of which is particularly unknown in many areas of the Southern Hemisphere. Our method offers the opportunity to implement a dynamic monitoring system using validated areas as a reference and can be used to detect other intertidal kelp areas in other marine ecoregions. The global map was developed using a Sentinel-2 dataset and does not consider differences in tidal heights. It may have errors caused by the elevation mask, since ALOS, DSM, and SRTM are not accurate at very fine scales (<30 m of spatial resolution) between land and sea. Further analyses will be done with a land mask of finer resolution when this becomes available. The tool created by [47] to clean up the effect of clouds was found to be a good option for making an averaged map of permanent cloud-free grid cells. However, permanent clouds could have covered some kelp forests for the whole period in some extreme environments.

## 5. Conclusions

The results of this study show that the use of the kelp filter algorithm presented here provides a simple, reliable, and inexpensive way of detecting and monitoring giant kelp forests covering areas larger than one hectare and in regions with tidal fluctuations < 4 m at different temporal and spatial scales. The method can be applied to the indices NDVI and FAI, but works best when applied to the KD. This system does not discriminate between giant kelp and green intertidal algae, but it discriminates between those signatures and other land and water grid ROIs. The filter algorithm was successfully validated with UAV imagery. Additionally, the platform provided by GEE provides pre-processed images and constantly updated datasets, and the code is open for application by any user.

This methodology provides a repeatable and robust process to assess the distribution and abundance of giant kelp forests, allowing this tool to contribute to the wider temporal monitoring of giant kelp communities. This dynamic approach would have the potential to address questions linked to their connectivity in their widespread extension in the world's ocean [7] and to kelp's global changes in distribution and abundance.

**Author Contributions:** Conceptualization, A.M.-S. and M.M.-F.; methodology, A.M.-S. and M.M.-F.; software, A.M.-S.; validation, M.P., A.M.-S., E.C.M., A.P.-M., N.G., M.Y. (Mary Young), P.H. and I.G.; formal analysis, A.M.-S.; investigation, A.M.-S.; resources, M.P., P.H., I.G., N.G.; data curation, M.T.; Writing—Original draft preparation, A.M.-S.; Writing—Review and editing, A.M.-S., M.P., E.C.M., A.P.-M., N.G., M.Y. (Mohammad Yaqub), P.H., I.G. and M.M.-F.; visualization, A.M.-S. and M.Y. (Mohammad Yaqub); supervision, M.-M.F.; project administration, A.M.-S.; funding acquisition, M.P., P.H., I.G., N.G., A.M.-S. All authors have read and agreed to the published version of the manuscript.

**Funding:** The fieldwork of this research was funded by CONICYT (scholarship 21171029 awarded to M. Palacios), Centro de Investigación de Ecosistemas Marinos de Altas Latitudes (FONDAP—IDEAL grant 15150003 from CONICYT), Centro de Investigación de Ecosistemas de la Patagonia (CIEP), South Atlantic Environmental Research Institute (SAERI), the School of Geography and the Environment—University of Oxford, St Peter's College (Graduate Awards) and Santander Academic Travel Awards. This research is part of A.M-S's PhD funded by CONICYT- Becas Chile. Drone imagery collection from the Falklands Islands was undertaken by Neil Golding at SAERI, as part of the Darwin DPLUS065 Coastal Habitat Mapping project, grant aided by the Darwin Initiative through UK Government funding.

**Acknowledgments:** We gratefully acknowledge the contribution to this publication made by the Falkland Islands IMS-GIS data centre. Thanks to Giovanni Daneri and Madeleine Hamamé (CIEP), Federico Betti (U. Genoa), Mathias Hüne and Fernando Luchsinger (Fundación Ictiológica), Pamela Fernández (i-Mar), Austin Capsey (UKHO) and Tom Hart (U. Oxford) for providing kelp data and commenting on the results. Thanks to IDEAL, CIEP, Américo Montiel (UMAG), Celestino Ancamil (Puerto Cisnes), Miguel Herrera (Maitencillo), SAERI and Shallow Marine Surveys Group (Falkland Is.) for their logistic help and field support. The constructive suggestions from three anonymous reviewers and the editors helped to improve the final manuscript. We are grateful to the colleagues that commented on the manuscript on earlier versions: Daniela Manuschevich (UAHC) and José Luis Iriarte (UACh). Finally, thanks to Noel Gorelick, Nicholas Clinton and David Carmichael of GEE for their classes and assistance.

**Conflicts of Interest:** The authors declare no conflict of interest. The funders had no role in the design of the study; in the collection, analysis, or interpretation of data; in the writing of the manuscript, or in the decision to publish the results.

## Appendix A

https://github.com/BiogeoscienceslabOxford/kelp_forests Kelp filter algorithm (JS code); box and whisker plot of NDVI, FAI and KD and their thresholds; methodology workflow; high-resolution validation (GEE interface); Drone-KD-CART and RF comparison; low-resolution kelp observations (table).

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
