# Peer review of "A High-Resolution Global Map of Giant Kelp (Macrocystis pyrifera) Forests and Intertidal Green Algae (Ulvophyceae) with Sentinel-2 Imagery"

_remotesensing, doi:10.3390/rs12040694_

Round 1
Reviewer 1 Report
The study presents very interesting results on the usage of satellite images to map giant kelp forests and to monitor their changes. The manuscript is well structured and provide all relevant information in a very substantitve manner. Overall, I highly recommend the manuscript for publishing and only have a few minor comments.
line 129: replace "last 3 years" by actual years (2015-18 as in line 141)
line 209/210 km without a capital k
line 217: UAV has not been explained before
Figure 6: Sentinel 1C might be a bit misleading - there should be enough space to insert Sentinel 2-L1C
line 355: it remains unclear to what "expected noise" refers to. Internannual variabilty within validation data? Or bad data quality? Or else? Why was it expected?
Reviewer 2 Report
The manuscript “A high-resolution global map of giant kelp forests and intertidal green algae with Sentinel-2 images” by Mora-Soto et al. develops global thresholds of remotely-sensed spectral values for Kelp and validates them with independently collected field data from the Chilean coast and the Falklands. They furthermore propose a Kelp-difference index that works by differentiating the red bands of Sentinel 2 Imagery.
Although the authors claim to have produced a ‘global map’ of giant kelp forests, their results and online interface show that they have in fact only mapped one particular kelp species ( Macrocystis pyrifera ) and furthermore only in selected regions. There are many other kelp species and ectomorphs that the authors do no consider, but which can be considered part of the global kelp biome. I therefore suggest they adjust the title and aims accordingly in order to not misled readers.
Parts of the manuscript has been poorly written with confusing terminology and description of previous approaches, particular so in the introduction. I recommend the authors spend some time to carefully flesh out what makes this study novel compared to previous studies.
The methods are quite convoluted and partially unclear. Furthermore the study could be much improved by using an algorithm capable of differentiating the varying classes and thresholds, such as for instance a classification tree (CART, Randomforest, … ), which are available in Google Earth Engine. Currently it reads as if the authors decided upon the used threshold based on visual inspection which seems rather subjective to me and is furthermore highly sensitive to noise. Despite overall impressive effort of this study, I would like to see that the authors how the authors justify their threshold compared to any data-driven approach?
Major issues:
L61ff: Here the authors list a number of satellites and their resolution while referring to previous studies. I suggest the author go into a bit more detail how these studies mapped kelp forest and not what satellite imagery they used! Similar as in the last paragraph. It is not clear to me what makes all these studies different, what the progress in mapping is or what methodologies work and which not. I suggest some major rewording and restructuring of these paragraphs, for instance by introducing first (1) the imagery data used and (2) what methodologies (algorithms) were used to map kelp forest.
L172: It is not clear to me why the authors discarded certain training data points just with the note that those ‘could be part of the commission error’? I am dubious to what the influence of these point is on the resulting map. From Figure 2 it appears as if many Kelp sites can have values above the excluded threshold (dotted line)?
L177-180: I don’t understand the reasoning for using NDVI as a spectral indicator for marine systems, especially since coastal cells and land vegetation is excluded a priori. There are many other indices particular for marine ecosystems, but see
Mouw, Colleen B., et al. "A consumer's guide to satellite remote sensing of multiple phytoplankton groups in the global ocean." Frontiers in Marine Science 4 (2017): 41. https://www.frontiersin.org/articles/10.3389/fmars.2017.00041/full
L194: As above: How were those valued identified as commission errors? Similar as L200 and Table 2. I think the authors need to objectively explain how these threshold were derived. Was there an analysis involved? Or was it just an visual assessment? There are multiple algorithms available – even within GEE – that could assist in deriving robust thresholds, such as for instance classification and regression trees (CART).
L363: Ensure that you use correct tenses in all sections of the manuscript.
Discussion: The authors list a number of environmental factors that could influence their results. Any modern classification algorithm would be able to incorporate those factors into the resulting classification.
Figure 1: Colours are not optimal (green to blue are hardly distinguishable )and figure differs in style from all other figure. Please adjust and decide on a single cohesive visualization style. The figure legend lists some abbreviations for classes but not all.
Figure 2: Different visual style than Figure 1. Adjust. Furthermore I suggest describing what the boxplots measure (are those tukey boxplots?) and also insert the sample size (number of grid cells) above or below each boxplot.
Figure 3: Is this a key figure? I see no obvious differences between the metrics, which raises for me more questions than insight. Maybe move to the Supporting Material.
Figure 4: State the projection used for this map. I was confused to see Chile upside-down?
Figure 6: Key figure? Move to Supporting Information if not. This is also in most parts already described in the text.
Table 5: I suggest highlight the estimates the indicator with the highest accuracy in bold. Nobody wants to search through large tables.
Figure 7: The shown values are continuous, therefore use a continuous legend for the maps, showing the range of possible values. Not 5 discrete bins.
Online interface: Overall very nicely designed, but a few improvements could be made. It was unclear to me how I would see the predicted Kelp forests as at large only the points are visible. Maybe the authors can provide some clearer instructions on the side to navigate users through the interface. Furthermore the thresholded KD values are best shown in a separate legend and/or when clicking on a coloured pixel.
Minor issues:
L28: Latin names and families in italic
L33: ‘the highest average kappa score’
L34: Don’t start sentences with a number, especially in the abstract. Consider rewording
L35: Unclear what is meant by ‘linear range’. Explain
L35: What algorithm specifically ? The rest of the abstract only introduces input data and predictors as well as accuracy scores
L42: Kelp forests ‘don’t’ increase richness. Species richness is a biodiversity measure. If at all, Kelp forest provide habitat for species
L46: ‘is a necessary step to understand… and identify …’
L47: Make a full stop here after the reference
L61: Survey of or for what? Specify
L84: Not the extraction, but the availability of endmember training data is crucial!
L87-88: Reword. ‘Probability increases … when few bands are used,’ ?
L94: Why were those studies not scaled up? Maybe provide potential reasoning here (Computational cost?)
L98: ‘Supervised classification’ is a very broad approach, be more specific and name the used algorithms (random forest, svms, .. ?)
L100: Reword. ‘Previous research, using citizen science data, found that … ‘
L101: What is an ‘automatic classification’ ?
L104-105: This comes out of nowhere and is confusing overall. What is a ‘temporal trend along the global distribution’ ? Why are temporal changes important? Very confused what the authors try to say here?
L108: Again, why is temporal replication necessary/important? These studies did not investigate change over time? This needs to be better communicated.
L114: Chlorophylls in lower case
L129: You mean an average composite
L165: It is unclear to me what the ‘count’ as summary statistic would indicate as the spectral values are continuous ?
L166: If coastal cells and land vegetation is masked out, why collect training data for it?
L168: Write out larger than or equal sign
L166-172: One generally mask’s areas ‘out’ and not ‘off’
L170: Unclear what this sentence tells us? What classes? How were the grid cells distributed? Suggest to alter Figure 2 (see above)
Generally refer to raster values as ‘grid cells’ and not ‘pixels’. A Pixel is what you see on your LCD screen.
L193: See issue above. What does the count quantify?
L218: You mean ‘Because of suboptimal weather conditions’ or similar
L225: Kelp no-kelp. ? This is not a full sentence. Delete here.
Table 3: If possible report coordinates with higher accuracy to make these sites more valuable for meta-analyses. Or alternatively provide the digitized polygons (Unless I have missed them).
L350: ‘need being highlighted’ ? Please correct the grammar in this sentence
L354ff: These are all guesses, can you demonstrate that this is the case? Otherwise I would counter that ‘a significant portion of the errors stem from using simplistic thresholding of spatially and temporally highly variable spectral indicators‘.
L360: ‘needs to be interpreted’
L376: Comma missing
Reviewer 3 Report
This manuscript tries to propose a study that uses Sentinel-2 to map Kelp coverage globally based on GEE. While there are sufficient validation which makes their results convincing, I still have some comments that might be helpful for the authors to reexamine their work.
It is not clear in the abstract that what does this algorithm in Line 35 refer to, NDIV? FAI? KD? Or a combination of these three.
According to the title, it seems that the authors tried to pay equal attention to kelp and green algae, but in the introduction, green algae are barely mentioned. This also exists in the Discussion, where there is a section of kelp mapping but no algae mapping. I suggest the authors to reorganize their thoughts and make the manuscript more consistent from the title to each single part.
I am not a biologist, I do not know if there are any connections or differences between kelp, giant kelp, algae, green algae. Would be helpful if there are some brief descriptions about this in the introduction.
I do not know if there is seasonality or phenology for kelp and algae. I suppose no, because you are using a composite sentinel-2 image for mapping their extent. And I also suppose their extent changes very slowly, because the temporal phase of the mapping is barely mentioned. Readers would benefit if you can add some sentences about this information into the discussion.
The employment of employing ALOS and SRTM for delineating ocean shoreline looks weird to me. I thought oceans can be easily extracted from Sentinel-2 images?
Figure 1 is not very clear to me. But it seems that GA is not distinguishable from ocean water or other classes, because their spectral characteristics are too similar. Then I would doubt how can GA be identified using NDVI or FAI or any other indices?
According to Figure 1, I cannot figure out why do you think B6 is better than B8? It seems a index of B8-B4 or just NDVI would outperform your KD index in extracting Kelp, especially considering that both B8 and B4 have the same resolution (10 m), while B6 has a resolution of 20 m. I do not understand why you tried to propose the KD index, from the description in your paper. Please convince me.
